

# Energetics and evasion dynamics of large predators and prey: pumas *vs.* hounds

Caleb M. Bryce[1,2], Christopher C. Wilmers[3] and Terrie M. Williams[1]

[1] Department of Ecology & Evolutionary Biology, University of California, Santa Cruz, CA, United States of America
[2] Botswana Predator Conservation Trust, Maun, Botswana
[3] Center for Integrated Spatial Research, Environmental Studies Department, University of California, Santa Cruz, CA, United States of America

Corresponding author
Caleb M. Bryce, cbryce@ucsc.edu, calebmbryce@gmail.com

## ABSTRACT

Quantification of fine-scale movement, performance, and energetics of hunting by large carnivores is critical for understanding the physiological underpinnings of trophic interactions. This is particularly challenging for wide-ranging terrestrial canid and felid predators, which can each affect ecosystem structure through distinct hunting modes. To compare free-ranging pursuit and escape performance from group-hunting and solitary predators in unprecedented detail, we calibrated and deployed accelerometer-GPS collars during predator-prey chase sequences using packs of hound dogs (*Canis lupus familiaris*, 26 kg, $n = 4$–5 per chase) pursuing simultaneously instrumented solitary pumas (*Puma concolor*, 60 kg, $n = 2$). We then reconstructed chase paths, speed and turning angle profiles, and energy demands for hounds and pumas to examine performance and physiological constraints associated with cursorial and cryptic hunting modes, respectively. Interaction dynamics revealed how pumas successfully utilized terrain (e.g., fleeing up steep, wooded hillsides) as well as evasive maneuvers (e.g., jumping into trees, running in figure-8 patterns) to increase their escape distance from the overall faster hounds (avg. 2.3× faster). These adaptive strategies were essential to evasion in light of the mean 1.6× higher mass-specific energetic costs of the chase for pumas compared to hounds (mean: 0.76 *vs.* 1.29 kJ kg$^{-1}$ min$^{-1}$, respectively). On an instantaneous basis, escapes were more costly for pumas, requiring exercise at $\geq$90% of predicted $\dot{V}O_{2\,MAX}$ and consuming as much energy per minute as approximately 5 min of active hunting. Our results demonstrate the marked investment of energy for evasion by a large, solitary carnivore and the advantage of dynamic maneuvers to postpone being overtaken by group-hunting canids.

# INTRODUCTION

Hunting modes in sympatric large carnivores have evolved and diversified, with members of the families *Felidae* and *Canidae* exhibiting nearly opposite prey detection and capture techniques characterized by cryptic ambushing or cursorial pursuit, respectively (Table 1). Gray wolves (*Canis lupus*), for example, often hunt cooperatively in packs (*Mech, 1970*; *Mech, Smith & MacNulty, 2015*) and rely on endurance pursuit (*Snow, 1985*; *Poole*

**Table 1** **Comparison of hunting mode divergence observed in large felids and canids.** Selected references for each topic (superscripts) are provided below.

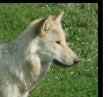

| | Family *Felidae* | Family *Canidae* |
|---|---|---|
| | **E.g., puma, leopard, jaguar** | **E.g., gray wolf, hound, dingo** |
| Hunting mode[a] | Cryptic stalking & pouncing; "Surprise & subdue" | Cursorial pursuit; "Charge & chase" |
| Hunting sociality[b] | Solitary | Often group/pack |
| Relative prey selectivity and timing[c] | Low (opportunistic); Prior to attack | High (selective); Often during pursuit |
| Interaction with & risk imposed by prey[d] | Short; Lower risk of injury/death | Prolonged; Higher risk of injury/death |
| Kill site attributes[e] | Sufficient structural cover for concealment during stalking and brief pursuit | Relatively open terrain that facilitates prolonged pursuit |
| Scale of habitat features impacting hunt success[f] | Small-scale habitat features | Large-scale landscape heterogeneity |
| Relative activity and energetic demand of hunt's attack phase | High intensity, short duration | Low intensity, long duration |

Notes.
[a] *Hornocker, 1970*; *Koehler & Hornocker, 1991*; *Ruth & Murphy, 2009a*; *Seidensticker et al., 1973*; *Young & Goldman, 1946*; *Poole & Erickson, 2011*; *Snow, 1985*; *Mech & Korb, 1978*; *Mech & Cluff, 2011*.
[b] *Gittleman, 1989*; *Hornocker & Negri, 2009*; *Mech, Smith & MacNulty, 2015*; *Mech, 1970*.
[c] *Husseman et al., 2003*; *Wilmers, Post & Hastings, 2007*; *Kunkel et al., 1999*; *Okarma et al., 1997*; *Mech, Smith & MacNulty, 2015*; *Peterson & Ciucci, 2003*; but see *Karanth & Sunquist, 1995*; *Krumm et al., 2010*.
[d] *Hornocker & Negri, 2009*; *Mech, Smith & MacNulty, 2015*; *Mech & Boitani, 2003*.
[e] *Alexander, Logan & Paquet, 2006*; *Hebblewhite & Merrill, 2008*; *Husseman et al., 2003*; *Ruth et al., 2011*; *Schmidt & Kuijper, 2015* and references therein.
[f] *Hebblewhite, Merrill & McDonald, 2005*; *Kauffman et al., 2007*; *Laundré & Hernández, 2003*; *Podgórski et al., 2008*; *Schmidt & Kuijper, 2015*.

*& Erickson, 2011*), rather than speed or agility, to test and ultimately outperform more vulnerable prey (*Peterson & Ciucci, 2003*; *Mech, Smith & MacNulty, 2015*). In contrast, pumas (*Puma concolor*) exhibit an opportunistic (i.e., less selective, *Husseman et al., 2003*; *Wilmers, Post & Hastings, 2007*), solitary and cryptic hunting mode by which they stealthily ambush and overpower prey (*Hornocker & Negri, 2009*; *Ruth & Murphy, 2009a*) through matching pounce force to prey size (*Williams et al., 2014*). Such divergence in locomotion, sociality, prey selectivity, and even preferred terrain while hunting reduces exploitative and interspecific competition (*Husseman et al., 2003*; *Elbroch et al., 2015*) through spatiotemporal niche partitioning and has cascading, ecosystem-wide effects (*Rosenzweig, 1966*; *Linnell & Strand, 2000*; *Donadio & Buskirk, 2006*; *Elbroch et al., 2015*). Less is known, however, of how the fine-scale movement, performance, and metabolic demands associated with these distinct predatory hunting modes interact to affect predation success.

During a predation event, an animal's ability to readily adjust its speed (*Howland, 1974*; *Domenici, 2001*), acceleration (*Combes et al., 2012*; *Wilson et al., 2013b*), and turn capacity (*Howland, 1974*; *Maresh et al., 2004*; *Wilson et al., 2013a*) becomes critical for survival. Despite its relative brevity, the attack phase of the hunt may be the most energetically expensive stage of prey acquisition, particularly for ambush predators (*Williams et al., 2014*). Given the two-dimensional confines of the terrestrial environment, both predators

and prey have restricted behavioral options during this critical phase and are primarily left with modulating their speed (*Elliott, Cowan & Holling, 1977*) and/or maneuverability (*Howland, 1974*) in order to hunt successfully or survive, respectively (reviewed in *Wilson et al., 2015*). Furthermore, these constraints may result in an ''arms race'' evolutionary escalation of matched, specialized morphologies and behavioral strategies that promote capture ability or evasion capacity in species that have co-evolved (*Brodie & Brodie, 1999*; *Cortez, 2011*), although the strength of selective forces acting on predators *vs.* prey may differ (i.e., the "life-dinner principle", *Dawkins & Krebs, 1979*).

The impacts of locomotor performance and energetics in altering chase outcomes has long been recognized, with the majority of our understanding of these interactions coming from studies of animals maneuvering in aerial (*Warrick, 1998*; *Hedenström & Rosén, 2001*; *Combes et al., 2012*) or aquatic (*Domenici & Blake, 1997*; *Domenici, 2001*) environments. Considerably less attention has been given to describing these complex dynamics in terrestrial species, particularly large carnivores and their prey (*Wilmers et al., 2015*). This is likely because our ability to describe such interactions is substantially impaired by the wide-ranging, often cryptic behaviors of these mammals (*Gese, 2001*; *Williams et al., 2014*; *Wang, Allen & Wilmers, 2015*). Recently, however, advancements and miniaturization of biologging sensor technology now enable scientists to concurrently measure previously unavailable metrics including the fine-scale behavior, physiological performance, and energetics of wild animals (*Kays et al., 2015*; *Wilmers et al., 2015*). In addition, these novel tools have the capacity to quantify chase dynamics and identify features of the landscape and the animals themselves that determine whether or not prey evade capture (*Wilson et al., 2013a*).

Here, using simultaneously instrumented pumas and scent hounds (*Canis lupus familiaris*), we examined the performance and energetic tradeoffs of divergent terrestrial hunting modes in real time. Packs of trained hounds pursued solitary pumas in need of recapture for a separate monitoring study and afforded a comprehensive look at hound group hunting cohesion and its effect on puma maximal performance escape tactics in rugged terrain. Given their local adaptation and stalk-and-pounce hunting mode, we predicted that pumas would exhibit greater acceleration, top speed, and turning ability in rugged terrain relative to hounds, but could only sustain this peak performance over a short distance and duration. Furthermore, we predicted that the cursorial hounds would compensate for poorer sprinting performance by coursing continually over long distances at slower speeds with greater energetic efficiency relative to pumas. Due to the scarcity of studies investigating detailed chase performance parameters and their associated metabolic costs in terrestrial mammals, our goal was to assess how terrain and evolved differences in morphology, physiology, and behavioral strategies among large felids and canids affect chase dynamics and outcomes.

## MATERIALS & METHODS

### Collar & energetic calibrations

We used a laboratory-to-field approach in which the locomotor biomechanics and energetics of scent hounds ($n = 7$, $24.2 \pm 0.9$ kg, mean $\pm$ SE) and captive pumas

($n = 3$, 65.7 $\pm$ 4.4 kg) instrumented with accelerometer-GPS collars were measured in an enclosure and laboratory environment prior to deployment on free-ranging conspecifics in the wild (*Bryce & Williams, 2017*; *Wang, Allen & Wilmers, 2015*; *Williams et al., 2014*; *Wilmers et al., 2015*). Hounds wore a 16 Hz accelerometer (TDR10-X, Wildlife Computers, Redmond, WA, USA) affixed to a GPS collar (Astro, Garmin Ltd, Switzerland) capable of taking a GPS satellite fix every 3 s (total collar mass = 328 g) and pumas wore an integrated accelerometer-GPS collar (GPS Plus, Vectronics Aerospace, Germany; total collar mass = 480 g) that sampled acceleration continuously at 32 Hz and took GPS fixes every 6 s in the field during hound-assisted puma recaptures. For both collar types, tri-axial accelerometer orientation was such that the $X$-, $Y$-, and $Z$- axes were parallel to the transverse, anterior-posterior, and the dorsal-ventral planes of the animal, respectively. For collar calibration, captive pumas (*Williams et al., 2014*; *Wang, Allen & Wilmers, 2015*) and hounds (*Bryce & Williams, 2017*) were filmed (Sony HDR-CX290/B, 1080 HD, 60 p) moving across a range of natural speeds (rest to 2 m s$^{-1}$ and 4.7 m s$^{-1}$, respectively) while on a treadmill enclosed by a metabolic chamber. Collar-derived accelerometer signatures were then correlated to gait-specific locomotor costs by simultaneously measuring oxygen consumption ($\dot{V}O_2$) and overall dynamic body acceleration (ODBA; *Qasem et al., 2012*; *Wilson et al., 2006*) of the animals during steady-state resting and treadmill running (*Williams et al., 2014*; *Bryce & Williams, 2017*). Because both speed and metabolic rate are linked to the dynamic component of an animal's body acceleration (*Gleiss, Wilson & Shepard, 2011*; *Bidder, Qasem & Wilson, 2012*; *Qasem et al., 2012*; *Bidder et al., 2012*), we used ODBA to translate sensor output from the collars into the speed, turning maneuvers, and energetics of free-ranging individuals.

## Fieldwork

An estimated population of 50–100 pumas resides in our 1,700 km$^2$ study area in the Santa Cruz Mountains of California (37°10.00′N, 122°3.00′W). The climate is Mediterranean, and elevation ranges from sea level to 1,155 m with rugged, forested canyons characterizing much of the preferred puma habitat. Human development (ranging from low-density to urban) is surrounded by native vegetation comprised of redwood and Douglas fir, oak woodland, coastal scrubland, and grassland communities. As a result, pumas and native mesopredators (i.e., coyotes, foxes, and bobcats) in the region exhibit spatial and temporal partitioning of activities that varies with human use (*Wang, Smith & Wilmers, in press*; *Wilmers et al., 2013*; *Smith, Wang & Wilmers, 2015*; *Smith, Wang & Wilmers, 2016*; *Wang, Allen & Wilmers, 2015*).

Previous work validated the use of accelerometer-GPS collars for describing spatiotemporally explicit puma energetics (*Wang, Smith & Wilmers, in press*; *Williams et al., 2014*) and behaviors (*Wang, Allen & Wilmers, 2015*) in the field. Here, we separately recaptured two adult male pumas (36 M and 26 M, 59.7 $\pm$ 0.7 kg) in autumn 2015 using packs of 5 and 4 hounds ($n = 8$, 24.3 $\pm$ 0.8 kg; *Wilmers et al., 2013*), respectively. We took advantage of this routine capture technique to simultaneously record and quantify the detailed chase-escape dynamics and associated energetic costs for hounds and pumas. In the field, we filmed (Sony HDR-CX290/B, 1080 HD, 60 p) the hound collar being manually

shaken prior to and immediately following deployment for subsequent accelerometer and GPS clock synchronization to Greenwich Mean Time (GMT). Similarly, we filmed the screen (including the clock) of the handheld UHF terminal (Vectronic Aerospace, Germany) while uploading the rapid GPS fix schedule to later synchronize the exact time of the puma collar schedule upload to that of the hound collar clock.

Puma chases occurred during daylight hours between 09:00 and 15:00 local time, a period that typically corresponds with inactivity for these nocturnal hunters (Fig. S1). Each puma initially escaped into terrain that precluded darting. As a result, we re-chased each puma after several hours and thus measured a total of four puma pursuits by hounds. All hounds were released simultaneously for each recapture, and although only one hound in each chase wore the combined accelerometer-GPS collar, all hounds wore identical GPS tracking collars to enable an analysis of hound pack hunting dynamics. After escaping to a tree suitable for darting, pumas were tranquilized with Telazol at a concentration of 100 mg/mL, measured, and re-collared while we collected the previous collar for chase reconstruction and analysis.

### Ethics statement

This study was conducted in strict accordance with animal ethics including capture and handling as approved by the California Department of Fish and Wildlife (Scientific Collection Permit #SC-11968) and the UC Santa Cruz Animal Care and Use Committee (IACUC Protocol #Wilmc1101). All human interventions including capture, administration of immobilizing drugs, radio collaring, monitoring were done to minimize negative/adverse impacts on the welfare of the study species.

### Analyses

From each chase, we quantified the speed, turning, energy expenditure, and elevation profile run by each instrumented animal. Instantaneous energetics and cost of transport (COT, the energy expended per meter) of pumas and hounds were determined by correlating 2 s smoothed ODBA (*Wilson et al., 2006*; *Shepard et al., 2008*), to laboratory-derived rates of oxygen consumption. We then used Eqn. (5) from *Williams et al. (2014)* to compare the COT of 60 kg pumas during typical 2-hour pre-kill active hunting activity (i.e., searching and stalking) to that of the brief, high intensity escape bouts during hound-assisted recapture. To assess the extent of anaerobic exercise for each species, we compared accelerometer-derived estimates of $\dot{V}O_2$ during chases to published values of $\dot{V}O_{2\,MAX}$ for lions (approx. 52 ml $O_2$ $kg^{-1}$ $min^{-1}$; *Taylor et al., 1980*; *Williams et al., 2014*) and dogs (approx. 160 ml $O_2$ $kg^{-1}$ $min^{-1}$, *Seeherman et al., 1981*; *Weibel et al., 1983*) of similar mass.

Overground pursuit and escape speeds were quantified by GPS-derived means for all animals, with accelerometer-derived speeds also computed for both pumas and focal hounds instrumented with combined sensors. The proportion of time spent not moving within each chase was calculated for each species based on the number of 2-second windows where ODBA <0.5 g. We downsampled all hound GPS data to fixes taken every 6 s to account for differences in GPS sample rate during chases and permit direct comparisons of hound and puma spatial datasets. The precise start and end of pursuits and escapes for hounds

and pumas, respectively, were determined by post-hoc comparison of GMT-synchronized video recordings obtained in the field and from each collar's raw accelerometer output. The beginning of each escape was readily apparent from puma accelerometer records, as each animal had been resting prior to hound release. Tag synchronization and data visualization was performed in Igor Pro (Wavemetrics, Lake Oswego, OR, USA). Statistical analyses and figures were produced using JMP Pro13 (SAS Institute Inc., Cary, NC, USA), program R (v. 3.1.1; *R Core Team, 2014*), and Matlab (Mathworks Inc, Natick, MA, USA). Study results are expressed as the mean $\pm$ SE ($\alpha = 0.05$, a priori).

## RESULTS

### Captive calibrations

For both hounds and pumas, mass-specific metabolic rate increased linearly as a function of ODBA as described previously for a variety of other terrestrial quadruped species (e.g., *Brown et al., 2013*; *Halsey et al., 2009*; *Wilmers et al., 2015*), according to

$$\dot{V}O_{2\ HOUND} = 22.87 \cdot ODBA + 6.39; \quad (r^2 = 0.86, n = 83, p < 0.001), \tag{1}$$

$$\dot{V}O_{2\ PUMA} = 58.42 \cdot ODBA + 3.52; \quad (r^2 = 0.97, n = 9, p < 0.001), \tag{2}$$

respectively, where $\dot{V}O_2$ is in ml $O_2$ kg$^{-1}$ min$^{-1}$ and ODBA is in g. Similarly, speed was strongly predicted from ODBA (*Bidder, Qasem & Wilson, 2012*; *Bidder et al., 2012*) for both species according to

$$Speed_{HOUND} = 2.56 \cdot ODBA - 0.32; \quad (r^2 = 0.82, n = 83, p < 0.001), \tag{3}$$

$$Speed_{PUMA} = 5.32 \cdot ODBA - 0.42; \quad (r^2 = 0.85, n = 9, p < 0.001) \tag{4}$$

where speed is in m s$^{-1}$ and ODBA is in g. Equations (2) and (4), as well as additional puma collar calibration data, are available from *Williams et al. (2014)* and *Wang et al. (2015)*.

### Chase reconstructions

The duration, distance, average speed, elevation change, and number of hounds involved in each recapture are summarized for hounds and pumas in Table 2. We present individual chase tracks and parameters (Figs. 1, S2–S4) as well as a Google Earth Pro (earth.google.com) visualization of chase 4 generated from synchronized puma and hound collar data (Video S1). In general, mean chase distance was three times farther for hounds (1,020 $\pm$ 249 m) than pumas (335 $\pm$ 63 m, $t(6) = -2.66$, $p = 0.037$, Table 2) because we released hounds from a distance great enough to not startle pumas prior to release. In this way, we measured the complete and varied escape maneuvers of the puma in response to the approaching hounds. As a result of these longer pursuit distances, hound chase duration (08:59 $\pm$ 03:05 mm:ss) was longer than the associated escape time in pumas (03:48 $\pm$ 01:16 mm:ss). Compared to the initial escape, each puma's second escape was shorter in both distance (247 $\pm$ 69 *vs.* 423 $\pm$ 60 m) and duration (01:53 $\pm$ 00:56 *vs.* 05:44 $\pm$ 01:12 mm:ss, Table 2).

As predicted, tortuosity (total distance traveled divided by straight-line distance from start to end point of run) did not differ significantly between hounds and pumas when

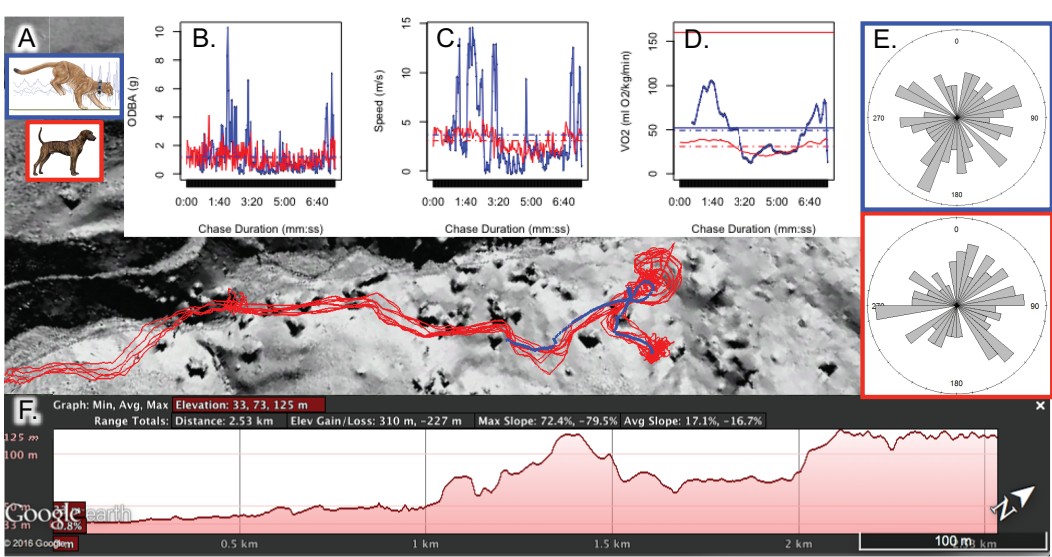

**Figure 1** **Chase 1 pursuit (red lines = hounds) and escape (blue line = puma) paths (A), including the elevation profile for Brandy, a GPS-accelerometer collar equipped hound.** Insets display ODBA (g, B), speed (m s$^{-1}$, C), and estimated mass-specific metabolic demand ($\dot{V}O_2$ in ml O$_2$ kg$^{-1}$ min$^{-1}$, D) For (B), (C), and (D), mean values are presented as dashed horizontal lines, and solid horizontal lines in (D) depict $\dot{V}O_{2\,MAX}$ for each species. Tortuosity plots (proportion of turns in each compass direction, (E) and the elevation profile for the accelerometer-GPS-equipped hound (F) are also presented. Map data© 2016 Google.

**Table 2** **Summary of pursuit and escape parameters from hounds and pumas, respectively.** Measurement units are enclosed in parentheses. Average speed (m s$^{-1}$) is GPS-rather than accelerometer-derived, and across-chase averages (±SE) are presented.

| | | Hound | | | | Puma | | | |
|---|---|---|---|---|---|---|---|---|---|
| | Hounds (n) | Distance (m) | Duration (mm:ss) | Avg. speed (m s$^{-1}$) | Elev. Gain/ Loss (m) | Distance (m) | Duration (mm:ss) | Avg. speed (m s$^{-1}$) | Elev. gain/loss (m) |
| Chase 1 | 5 | 1,270 | 07:37 | 2.78 | 228/−161 | 482 | 06:56 | 1.16 | 121/−84 |
| Chase 2 | 5 | 1,400 | 15:13 | 1.53 | 306/−157 | 178 | 02:48 | 1.06 | 70/−32 |
| Chase 3 | 4 | 1,120 | 12:08 | 1.54 | 99/−339 | 363 | 04:15 | 1.33 | 88/−139 |
| Chase 4 | 4 | 291 | 00:59 | 4.93 | 80/−75 | 316 | 00:57 | 5.54 | 89/−110 |
| Average | | 1,020 | 08:39 | 2.7 | 178 (54) | 334.8 | 03:44 | 2.27 | 92 (11) |
| | | (250) | (03:05) | (0.8) | −183 (56) | (62.8) | (01:15) | (1.09) | −91 (23) |

all individuals and chases were grouped ($t$ (22): 1.04, $p = 0.31$; Table 3) because the scent hounds roughly followed each puma's escape path. Overground distance traveled averaged 2.3 to 2.9 times farther than straight-line distance, indicating extent of turning maneuvers while running through rugged terrain. To prolong the time until captured, pumas employed several adaptive strategies that compensated for physiological constraints and being outnumbered. Evasive maneuvers such as temporarily jumping into trees, running hairpin turns or figure-of-8 patterns, and fleeing up steep, wooded hillsides were all used repeatedly to increase escape distance and postpone being overtaken (Table S1). For example, when the hounds were within 35 m of puma 36 M (chase 1), the puma ran

**Table 3 Average (±SE) chase tortuosity and speed performance during hound-assisted puma recaptures.** Average and maximum speeds (m s$^{-1}$) are presented for both GPS and accelerometer-derived estimates. Sample sizes and measurement units are enclosed in parentheses, and results from Welch two-sample $t$-tests comparing hound and puma data are included.

| Chase | Species (animals) | Path tortuosity | GPS speed (m s$^{-1}$) | | Accel. speed (m s$^{-1}$) | |
|---|---|---|---|---|---|---|
| | | | Avg. | Max. | Avg. | Max. |
| 1 | Hounds ($n=5$) | $2.01 \pm 0.08$ | $2.33 \pm 0.09$ | $8.53 \pm 0.29$ | $3.07 \pm 0.07$ | 5.2 |
| | Puma 36 M | 2.22 | $0.93 \pm 0.21$ | 5.27 | $3.69 \pm 0.27$ | 14.5 |
| | | | $t = -6.5$ | | $t = 2.3$ | |
| | | | $p < 0.01^*$ | | $p = 0.02^*$ | |
| 2 | Hounds ($n=5$) | $3.61 \pm 0.21$ | $1.53 \pm 0.03$ | $7.5 \pm 0.45$ | $2.43 \pm 0.05$ | 5.93 |
| | Puma 36 M | 3.34 | $0.56 \pm 0.17$ | 2.89 | $3.49 \pm 0.34$ | 15.0 |
| | | | $t = -6.4$ | | $t = 3.1$ | |
| | | | $p < 0.01^*$ | | $p < 0.01^*$ | |
| 3 | Hounds ($n=4$) | $1.98 \pm 0.09$ | $1.35 \pm 0.04$ | $5.5 \pm 0.87$ | $3.32 \pm 0.06$ | 6.4 |
| | Puma 26 M | 4.95 | $0.48 \pm 0.11$ | 2.38 | $2.85 \pm 0.2$ | 11.8 |
| | | | $t = -8.7$ | | $t = -2.2$ | |
| | | | $p < 0.01^*$ | | $p = 0.03^*$ | |
| 4 | Hounds ($n=4$) | $1.42 \pm 0.06$ | $3.04 \pm 0.2$ | $5.89 \pm 0.45$ | $5.35 \pm 0.15$ | 6.35 |
| | Puma 26 M | 1.15 | $2.32 \pm 0.51$ | 3.86 | $11.06 \pm 0.5$ | 14.49 |
| | | | $t = -1.4$ | | $t = 12.1$ | |
| | | | $p = 0.16$ | | $p < 0.01^*$ | |
| Avg. | Hound | $2.32 \pm 0.24$ | $1.7 \pm 0.03$ | $7.0 \pm 0.38$ | $3.89 \pm 0.18$ | $5.97 \pm 0.28$ |
| | Puma | $2.92 \pm 0.52$ | $0.74 \pm 0.09$ | $3.6 \pm 0.63$ | $2.94 \pm 0.04$ | $13.9 \pm 0.73$ |
| | | $t = 1.04$ | $t = -10.4$ | $t = -3.9$ | $t = 5.2$ | $t = 10.3$ |
| | | $p = 0.31$ | $p < 0.01^*$ | $p < 0.01^*$ | $p < 0.01^*$ | $p < 0.01^*$ |

**Notes.**
*Denotes significant relationship ($p \leq 0.05$). GPS speeds are inherently averaged over 6 s, whereas the accelerometer speeds are near instantaneous (see methods).

a figure-of-8 pattern and briefly jumped into a tree. As a result, hound-puma separation distance increased by nearly 15 m, and the puma's capture was delayed by an additional 5 min (Fig. S5).

Overall average chase speed, as measured by chase distance divided by chase duration, was similar between species ($2.7 \pm 0.8$ m s$^{-1}$ and $2.3 \pm 1.09$ m s$^{-1}$ for hounds and pumas, respectively; $t(6) = -0.31$, $p = 0.77$, Table 2). However, GPS-derived average speeds from all hounds (including those not outfitted with accelerometers) and pumas suggested that, across chases, hound pursuit speed was twice that of the escaping pumas (1.7 *vs.* 0.74 m s$^{-1}$ for hounds and pumas, respectively; $t(1870) = -10.4$, $p < 0.01$; Table 3, Fig. 2), since pumas spent larger proportions of each encounter stationary (avg. 31% *vs.* 15% stationary for pumas and hounds, respectively; $t(6) = 1.37$, $p = 0.22$). Using accelerometer-derived speed estimates (Eqs. (3) and (4)) to resolve running dynamics in finer temporal resolution, we note that pumas briefly hit peak speeds in excess of 14m s$^{-1}$, more than twice the top speed of the pursuing hounds (5.2–6.3 m s$^{-1}$, Table 3). Puma escapes were characterized by sequential high-speed evasive maneuvers interspersed with slow, low acceleration periods (Fig. 3). In contrast, hound pursuit speeds were more constant over the course of each chase (Fig. 1C and Figs. S2–S4C).

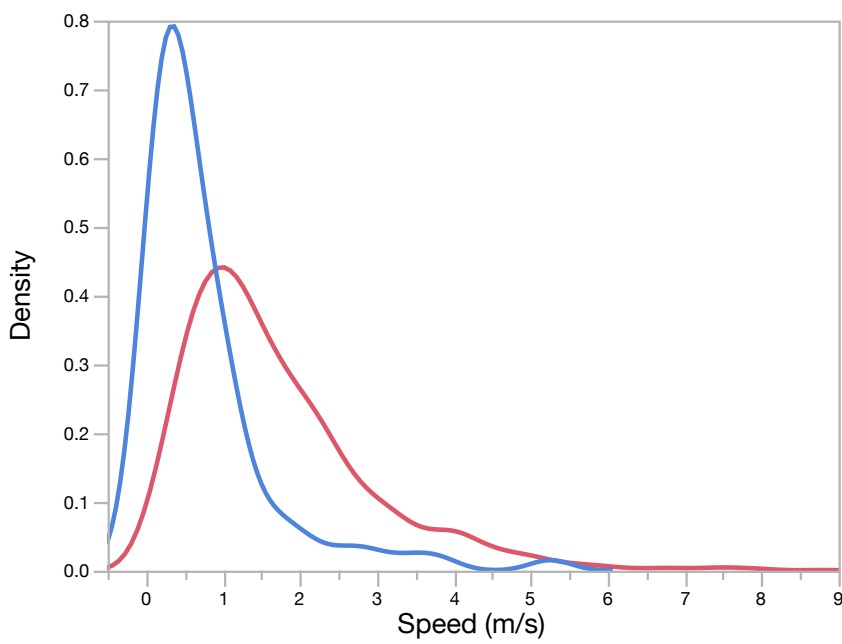

**Figure 2  GPS-derived pursuit and escape speeds for hounds (red) and pumas (blue), respectively, during all chases.** The mean ($\pm$SE) speeds, in m/s, for hounds ($1.7 \pm 0.03$) and pumas ($0.74 \pm 0.09$) are depicted as dashed vertical lines.

Immediately after release, all hounds concurrently worked to detect the nearby puma's scent and give chase. Average hound chase speed did not differ across individuals for both pursuits of puma 36 M (chase 1: $2.52 \pm 0.07$ m s$^{-1}$, $F_{4,711}$: 0.71, $p = 0.59$; chase 2: $1.74 \pm 0.03$ m s$^{-1}$, $F_{4, 1,653}$: 1.95, $p = 0.1$), but Hound 4 (Crocket) was significantly slower than the three other hounds during both pursuits of puma 26 M (Hound 4: $1.37 \pm 0.06$ m s$^{-1}$, others: $1.71 \pm 0.04$ m s$^{-1}$; $t_{988}$: $-4.4$, $p < 0.001$). This was probably a result of Hound 4's age (11), over twice as old as the other hounds (average age of 5) involved in 26 M's recapture. Hound group cohesion (the spacing of individual members in proximity to the moving group centroid, measured every 3 s) varied across chases (Fig. 4), likely due to interacting effects including pack composition, individual characteristics (e.g., experience, age, sex), topographic complexity, and puma scent freshness. Tighter spatial clustering was observed between the five members of the hound pack pursuing puma 36 M (Figs. 4A and 4B) than that of the 4-member pack that chased puma 26 M (Figs. 4C and 4D). Across chases, the maximum path deviation of individual hounds from the centroid of the moving group averaged 13.1 ($\pm 2.8$) meters. No single hound was ever beyond 55 m of the true path of the puma, although the average maximum deviation was 19.1 $\pm 11.7$ m.

### Energetic demands

Across chases, the metabolic rates (kJ min$^{-1}$) of pumas during escape ($76.5 \pm 15.1$) were nearly four times higher than those of the pursuing hounds ($20.1 \pm 15.1$, $t(6) = 2.65$, $p = 0.038$; Fig. 5). On a mass-specific basis, metabolic rates (kJ kg min$^{-1}$) were still $>1.6\times$ greater in pumas relative to hounds ($1.29 \pm 0.27$ *vs.* $0.76 \pm 0.27$ kJ kg min$^{-1}$). Similarly,

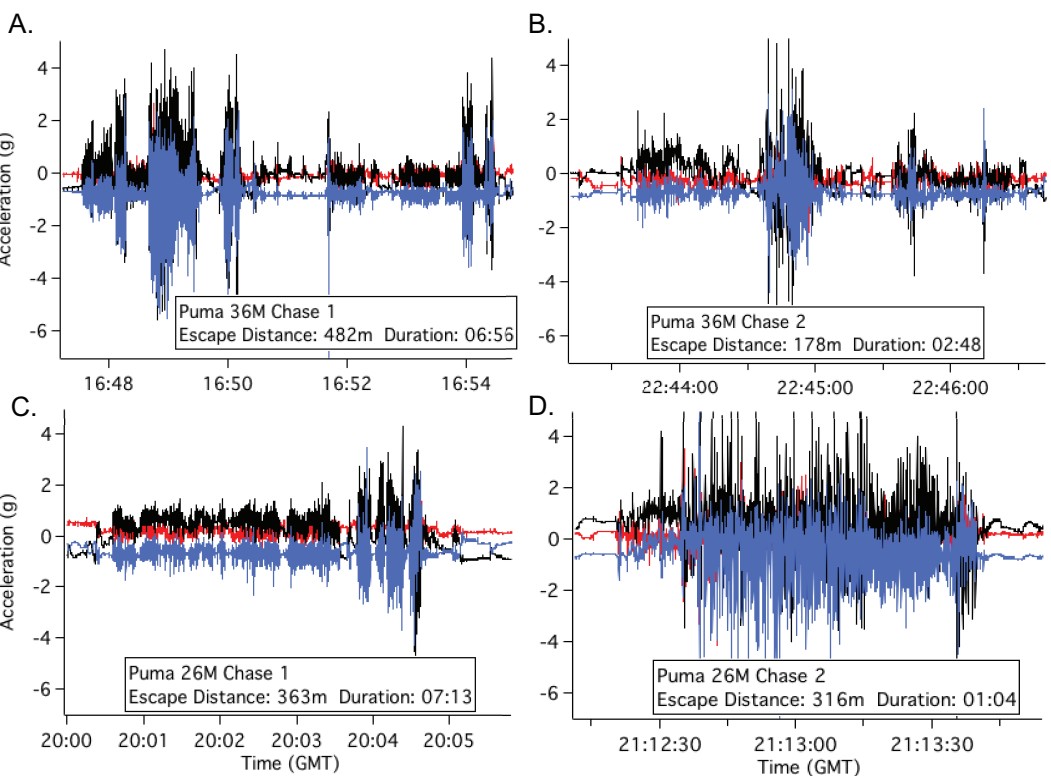

**Figure 3** **Escape acceleration signatures of adult male pumas 36 (A, B) and 26 (C, D).** Acceleration (g) is scaled to the same range for comparison. Chase distance is in m and chase duration is in mm:ss. Colors correspond to pumas' accelerometer-GPS collar orientation in the $X$ (transverse sway, black), $Y$ (anterior-posterior surge, blue), and $Z$ (dorsal-ventral heave, red) planes.

COT ($J \ kg^{-1} \ m^{-1}$) was $>2\times$ as high for pumas ($11.7 \pm 1.4$) than hounds ($5.5 \pm 1.4$, $t(6) = 3.05$, $p = 0.023$; Fig. 5).

Hounds remained below their gas exchange threshold (i.e., $\dot{V}O_{2\,MAX}$) for the duration of pursuits, with peak hound $\dot{V}O_2$ estimates during the highest-intensity chase (chase 4) of 60 ml $O_2 \ kg^{-1} \ min^{-1}$, or just 40% of $\dot{V}O_{2\,MAX}$ (Fig. 6). In contrast, pumas routinely exceeded $\dot{V}O_{2\,MAX}$ during escapes, with an average of $52.5 \pm 16\%$ (range: 32–100%) of each escape requiring energy from anaerobic metabolism.

Exercise effort was comparatively larger for pumas compared to hounds (Figs. 1D, 6 and Figs. S2–S4D) and on average, one minute spent escaping cost pumas $4.64 \times (\pm 1.3)$ as much energy as a typical minute spent actively hunting. In other words, the average puma escape duration of 03:48 ($\pm$01:16) was metabolically equivalent to about 18 min of routine, active hunting.

## DISCUSSION

In quantifying the fine-scale pursuit and evasion dynamics of two large carnivores, including free-ranging, cryptic pumas, we present evidence for morphological and physiological constraints imposed by specialization towards divergent hunting modes. Although the highly cursorial, endurance-adapted canids (here, scent hounds) exhibited relatively poor

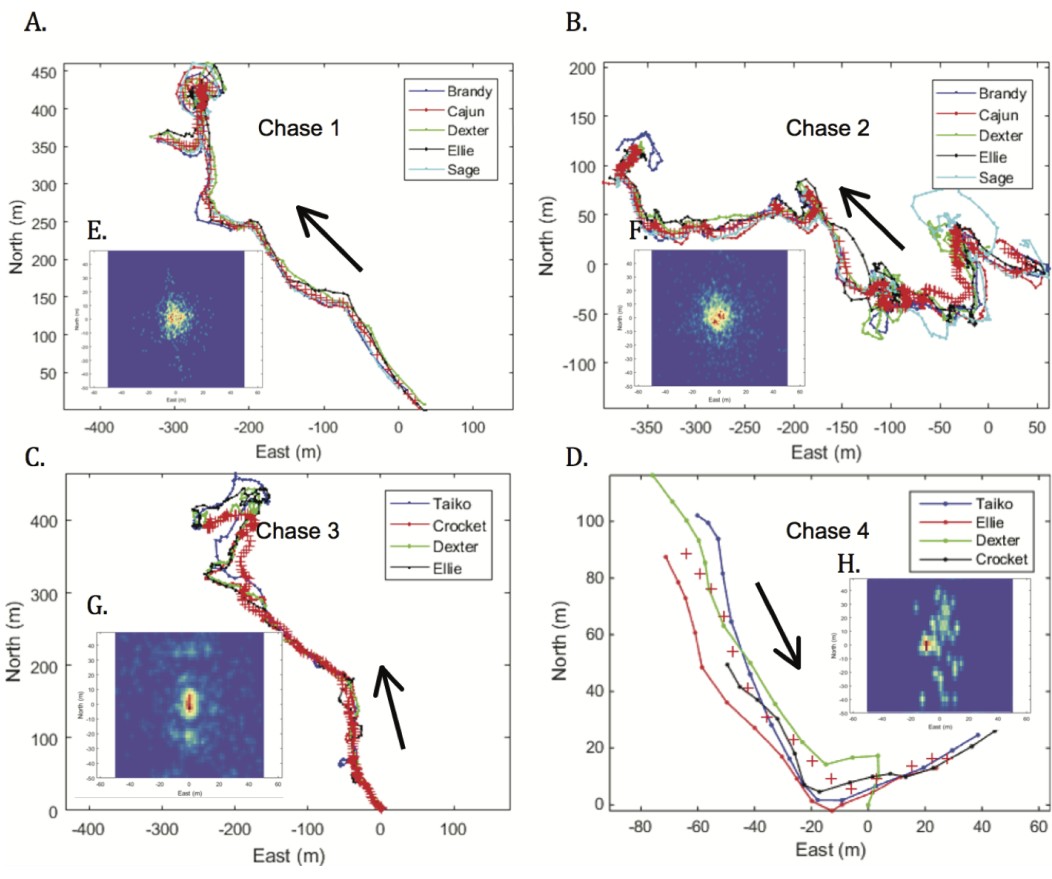

**Figure 4** **Hound pursuit paths (A–D) and 2D-spatial histograms (E–H) of pack cohesion over the course of each chase, with black arrows indicating direction of chase.** Every 3 s, the group centroid throughout each pursuit path is marked as a red plus (+). In the spatial histogram insets, the relative position of each hound relative to the group centroid is scaled by color, with warm colors representing close group cohesion and cool colors depicting more distant spacing.

turning ability and maximum speed compared with pumas, these animals maintained lower metabolic rates (Fig. 5B) and transport costs (Fig. 5D) than their felid quarry. Canids have a higher index of aerobic athleticism (*Gillooly, Gomez & Mavrodiev, 2017*; *Taylor et al., 1987*; *Weibel et al., 1983*), thanks to relatively larger hearts (*Williams et al., 2015*) and greater lung volumes (*Kreeger, 2003*; *Murphy & Ruth, 2009*) compared with felids of similar size. Like other endurance-adapted species (e.g., humans, horses), during high-intensity muscular exercise, canids likely exhibit higher critical speed/power and lower energy storage ($W'$) than felids (*Jones et al., 2010*; *Poole et al., 2016*). Canid skeletal specializations include 'box-like' elbow joints and limbs locked in a more prone position (*Figueirido et al., 2015*), enabling wolves, for example, to travel for several kilometers at 56–64 km hr$^{-1}$ (*Mech, 1970*; *Mech, Smith & MacNulty, 2015*), pursue prey over distances in excess of 20 km (*Mech & Korb, 1978*; *Mech, Smith & MacNulty, 2015*), and cover 76 km in 12 h (*Mech & Cluff, 2011*). As with other social canids, hounds worked together effectively as a pack (Fig. 4) to detect and maintain each puma's scent while giving chase through steep terrain and dense brush understory. Our results indicate that group-hunting hounds exhibit fission–fusion

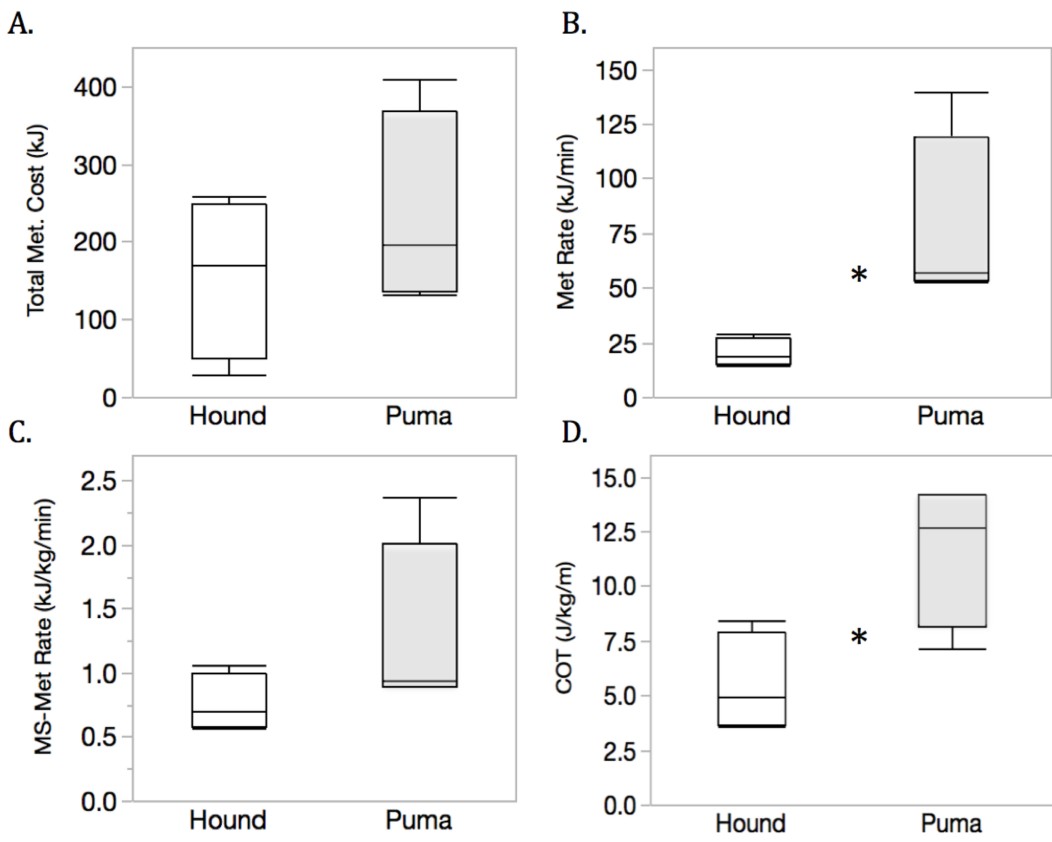

**Figure 5** **Energetic costs of pursuit and evasion for hounds (white) and pumas (grey), respectively, summarized across four chases.** Total metabolic cost (kJ, A), metabolic rate (kJ min$^{-1}$, B), mass-specific metabolic rate (kJ kg$^{-1}$ min$^{-1}$, C), and cost of transport (COT, J kg$^{-1}$ m$^{-1}$, D) are shown. Asterisks (*) denote significant ($p \leq 0.05$) differences between species.

spatial dynamics while giving chase. Furthermore, they suggest that pack size as well as the age, sex, and experience level of individuals can influence these complex dynamics. These and many other factors have been shown to impact group composition and kill success in group-hunting predators (reviewed in *Gittleman, 1989*).

In contrast, as solitary, highly adapted stalk-and-pounce predators, pumas rely heavily on an element of surprise coupled with a short pursuit ($\leq$10 m, *Laundré & Hernández, 2003*) before making contact with and subduing prey (*Murphy & Ruth, 2009*). Compared to canids, in felids, pouncing and grappling with prey are aided by wider elbow joints (*Figueirido et al., 2015*), greater spinal flexibility (*Spoor & Badoux, 1988*; *Ruben, 2010*), and other limb and pelvic adaptations (*Taylor, 1989*). We documented the extreme performance capabilities of this ambush-hunting mode (Table 3 and Fig. 3; also see *Williams et al., 2014*) as well as the physiological limitations for stamina exacted during the flight response (Figs. 5 and 6). For example, although brief, the maximum puma escape G-force measurements (Fig. 3) in excess of ±5 g are similar to those experienced during an Olympic luge race or under maximum braking force in a Formula 1 racecar (*Gforces.net, 2010*). Such peak performance capacity is energetically expensive (Figs. 1D, 6, S2D–S4D),

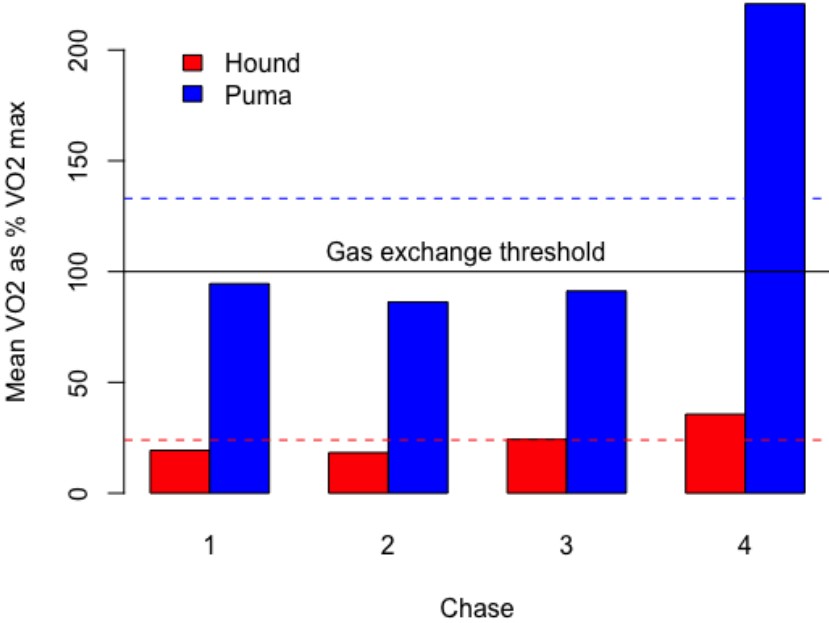

**Figure 6** Estimated mean metabolic rate ($\dot{V}O_2$, ml $O_2$ kg$^{-1}$ min$^{-1}$) expressed as a percentage of $\dot{V}O_{2\,MAX}$ for hounds (red) and pumas (blue) during each chase.

and each pumas' second escape was much shorter in distance and duration. In escape 4, for example, puma 26 M's brief burst of speed (Tables 2 and 3; Fig. S4) required short-term energy stores well beyond those met aerobically (i.e., above the gas-exchange threshold; Fig. 6). Our recapture results provide field-based empirical support for the locomotor ramifications of these hunting mode differences between large canids and felids.

Describing species-specific energetic costs and movement ecology can elucidate population-level consequences of anthropogenic disturbance and environmental change (*Stephens & Krebs, 1986*; *Gorman et al., 1998*; *Wikelski & Cooke, 2006*; *Somero, 2011*; *Seebacher & Franklin, 2012*; *Cooke et al., 2013*; *Humphries & McCann, 2014*; *Tomlinson et al., 2014*; *Laundré, 2014*; *Scantlebury et al., 2014*; *Wong & Candolin, 2014*). Cumulative costs associated with exposure to disturbance can tip the energy balance for large carnivores and potentially lead to demographic changes that reverberate through the ecological community (*Ripple et al., 2014*; *New et al., 2014*; *King et al., 2015*).

A recent terrestrial predator–prey pursuit model developed by *Wilson et al. (2015)* suggested that during predation interactions, the larger animal would be absolutely faster, but have inferior turning ability. Using our GPS-accelerometer datasets, we found greater maximum speeds and turning performance in pumas (weighing over twice the mass of each hound), but slower average speeds. Differences between our findings and predictions of the Wilson et al. model can be explained in part by recognizing our study's violation of several underlying model assumptions. For instance, our recaptures did not occur between a solitary predator pursuing solitary prey on flat and homogenous terrain, nor were the predator(s) and prey morphologically similar. In addition, some differences may

be explained by anatomical and physiological specialization in canids and felids, as well as local adaptations to rugged topography enhancing momentary escape performance in pumas.

We recognize that our opportunistic hound-assisted puma recaptures constitute semi-natural interactions, but nevertheless their analysis serves as an important first step in understanding the complexities and tradeoffs of locomotor performance *vs.* energy expenditure in large felids and canids (*Hubel et al., 2016*). For example, recaptures also enabled us to record the maximal or near-maximal performance capacity of a wild felid predator, shedding light on hunting adaptations for, and limits to, managing speed, maneuverability, and energy demand during prey capture. Our approach also provides a framework for quantifying natural competitive or predatory interactions and their outcomes in the future. Although dogs can adversely affect free-ranging carnivore behavior (e.g., *Vanak et al., 2013*; *Wierzbowska et al., 2016*), hound-elicited escapes by pumas may reflect direct interspecific interactions between wolf packs and solitary pumas in sympatric landscapes. While some evidence suggests that pumas are capable of killing subadult (*Ruth & Murphy, 2009b*) and adult wolves (*Schmidt & Gunson, 1985*), the pack hunting strategy of wolves generally makes them dominant competitors against solitary pumas during direct conflicts (*Husseman et al., 2003*; *Kortello, Hurd & Murray, 2007*; *Ruth & Murphy, 2009b*; *Ruth et al., 2011*; *Bartnick et al., 2013*).

Where they coexist, wolves and pumas often exhibit temporal as well as spatial niche partitioning, with pumas often utilizing edge habitat (*Laundré & Hernández, 2003*) or rugged terrain (e.g., steep slopes, boulders) dominated by vegetative cover for concealment when hunting (*Logan & Irwin, 1985*; *Laing & Lindzey, 1991*; *Williams, McCarthy & Picton, 1995*; *Jalkotzy, Ross & Wierzchowski, 2002*; *Husseman et al., 2003*). Wolves, in comparison, tend to prefer valley bottoms and open country for hunting (*Husseman et al., 2003*; *Alexander, Logan & Paquet, 2006*; *Atwood, Gese & Kunkel, 2007*; *Kortello, Hurd & Murray, 2007*). In addition to being critical for hunting cover (*Kleiman & Eisenberg, 1973*), pumas and other solitary felids rely on structural complexity and vegetative cover as escape terrain during direct intra- and interspecific conflict (*Duke, 2001*; *Ruth, 2004*; *Dickson, Jenness & Beier, 2005*; *Kortello, Hurd & Murray, 2007*). Furthermore, trees may serve as primary and immediate refuge from wolves and other threats as pumas do not readily utilize trees for other purposes, such as arboreal prey caching and consumption observed in other felids (e.g., lynx, leopards; *Balme et al., 2017*; *Vander Waal, 1990*). We observed this phenomenon in the field; each puma escape was characterized by agile, high-performance maneuvering that terminated with jumping into a tree immediately prior to being overtaken by the hound pack. Maintaining adequate vegetative cover therefore provides a dual concealment-safety benefit to pumas, indicating the importance of protecting complex habitat, in addition to adequate prey, to ensure the long-term persistence of these cryptic predators (*Beier, 2009*; *Burdett et al., 2010*; *Laundré, 2014*; *Williams et al., 2014*; *Wilmers et al., 2013*), especially where they co-occur with wolves (*Bartnick et al., 2013*; *Kortello, Hurd & Murray, 2007*; *Ruth & Murphy, 2009a*).

## ACKNOWLEDGEMENTS

We thank the California Department of Fish and Game, P Houghtaling, D Tichenor, T Collinsworth, and several undergraduate assistants for their significant field support during puma recaptures. We also thank JA Estes, O Dewhirst, L McHuron, B Nickel, and P Raimondi for their thoughtful discussions and contributions to the analyses.

### Funding

Support was provided by the National Science Foundation (DBI-0963022, DBI-1255913, and GK-12 DGE-0947923) and the Gordon and Betty Moore Foundation. Additional support came to CMB from the UCSC Science Internship Program (SIP), Mazamas, the ARCS Foundation, and the Ecology and Evolutionary Biology (EEB) Department at UC Santa Cruz. The funders had no role in study design, data collection and analysis, decision to publish, or preparation of the manuscript.

### Grant Disclosures

The following grant information was disclosed by the authors:
National Science Foundation: DBI-0963022, DBI-1255913, GK-12 DGE-0947923.
Gordon and Betty Moore Foundation.
UCSC Science Internship Program (SIP).
Mazamas.
ARCS Foundation.
Ecology and Evolutionary Biology (EEB).

### Competing Interests

The authors declare there are no competing interests.

### Author Contributions

- Caleb M. Bryce conceived and designed the experiments, performed the experiments, analyzed the data, contributed reagents/materials/analysis tools, wrote the paper, prepared figures and/or tables, reviewed drafts of the paper.
- Christopher C. Wilmers conceived and designed the experiments, contributed reagents/materials/analysis tools, reviewed drafts of the paper.
- Terrie M. Williams conceived and designed the experiments, reviewed drafts of the paper.

### Animal Ethics

The following information was supplied relating to ethical approvals (i.e., approving body and any reference numbers):

This study was conducted in strict accordance with animal ethics including capture and handling as approved by the the UC Santa Cruz Animal Care and Use Committee.

![PeerJ]

### Field Study Permissions

The following information was supplied relating to field study approvals (i.e., approving body and any reference numbers):

This study was conducted in strict accordance with animal ethics including capture and handling as approved by the California Department of Fish and Wildlife (Permit No. SC-11968).

### Data Availability

The raw data is provided in the Supplemental Files.

### Supplemental Information

Supplemental information for this article can be found online at http://dx.doi.org/10.7717/peerj.3701#supplemental-information.

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
