# Peer review of "Energetics and evasion dynamics of large predators and prey: pumas vs. hounds"

_PeerJ, doi:10.7717/peerj.3701_

## Round 0.1 · original submission · Minor Revisions

Both reviewers found this a very interesting study combining a valuable and original contribution to the field with a well written presentation. I concur with this view and am therefore pleased to accept it, contingent on minor revision to address the reviewers' concerns and some additional suggestions of mine.

Editor's Suggestions/Comments

L24. Provide scientific names at first mention in Abstract (already done in main text)
L55. I think this sentence refers to prey, but it is a bit ambiguous; it might be useful to be explicit.
L90. Intermittent locomotion often refers to a pattern of animal movement in which pauses, often very short, alternate with moves (see for example Kramer & McLaughlin 2001 Am Zool 41:137-153). It doesn't seem that this is what you intend here.
L100. At first use, define the error as SD or SE.
L129. If 'partitioning' is the subject, the verb should be 'varies'.
L168. Here and elsewhere, shouldn't the dot be above the V?
L174 'number of 2-second windows' (missing word)
L175. 6 s (insert space)
L274-276. This sentence is not very clear. Do you mean that the index of athleticism is higher in canids?
L397. Please check all your references for consistent format. Most scientific names need to be in italics. There is inconsistency in the capitalization of words in chapter titles (usually lower case except the first word and following a colon) and edited volume titles (usually all caps). A few journal titles are not capitalized. Are Ruben 2010 and Ruth 2004 doctoral theses or some other university reports? Please specify so that a reader would be able to locate them, if needed.

·

Basic reporting

A few minor concerns as below in author comments.

Experimental design

Major concern is making it clear up front that only 2 pumas were studied.

Validity of the findings

A few problems as identified below. These are perceived as correctable in revision.

Additional comments

GENERAL COMMENTS
This is an original and interesting paper that addresses a distinct knowledge gap. Overall the writing is clear and well organized, the design is appropriate (given the extremely challenging nature of the environment and species interactions studied) and the data are presented effectively. This work constitutes a lovely “next step” in approaching the “locomotor performance vs. energy expenditure” relationship in large felids and canids. Criticisms do not challenge the validity of the findings or the conclusions, as stated. Principal concerns relate to extrapolations based upon unsound physiological reasoning and inappropriate terminology (e.g. “anaerobic threshold”) and errors regarding energetics. These are perceived as readily addressable in revision.
SPECIFIC COMMENTS
LINE(S)
Introduction. In human exercise physiology (as well as horse, rodent and other quadrupeds) the concept of “Critical Power” or “Critical Speed”) is gaining traction to explain their exercise energetics and tolerance (see Poole and Erickson, 2011 (cited) as well as Jones et al. (Med. Sci.Sports Exerc. 42: 1876-90, 2010) and Copp et al. (J. Physiol. (Lond.) 588: 5077-87, 2011). Indeed,
The discrimination between sprinters (high W’, energy storage component, low critical speed) and distance (low W’, high critical speed) athletes would fit with the puma versus dog comparison herein.
Methods. Despite the heroic and very advanced data collection techniques and the perceived high quality data presented the n=2 pumas is a limitation that needs to be stated up front in the Abstract.
209 “mm:ss”
167 The term “anaerobic threshold” is now recognized as mechanistically in error. Specifically, nowhere has anaerobiosis been identified in skeletal muscle when lactate is being produced during exercise. Suggest the term “lactate threshold” or “gas exchange threshold” to depict how this metabolic rate was determined, as relevant.
170 Here and throughout. Replace “comparable” with “similar.” Anything can be co pared but it does not mean they are similar!
257 This carry over from the Weibel/Taylor days is simply not correct. Animals and humans increase their blood lactate concentration typically at 50-60% of their VO2max. At those metabolic rates there is no obligatory anaerobiosis. This well know in the human literature and, again, is obvious also from horses (e.g., McDonough et al. JAPPL 92: 1499-1505, 2002), dogs, rodents, and all other species studied – to my knowledge.
266- Again discussion of Critical Speed would be valuable here.
295 “exhaustion and/or overheating” Is there evidence collected herein to support this or is it speculation? If the latter, please remove.
Table 2. A better job could be done of explaining the large variability in average speed for dogs and pumas between chases.
Figure 5 legend. Please explain explicitly here the calculations and assumptions of moving from total metabolic cost to metabolic rate (i.e., B vs. A).

Reviewer 2 ·

Basic reporting

The basic reporting is clear and unambiguous, bar a few areas where clarity is sort (see below), often through expansion of the explanations provided. Very well references. Very professionally prepared. Results all relevant, and set up by a priori hypotheses.

L90: ‘but could only sustain this peak performance over a short distance and duration (i.e. more intermittent locomotion).’ – this points needs unpacking as it isn’t quite clear.

L107: Half a kilo of collar? That seems heavy to me. Any thoughts on whether it impacted on the animals?

L111-114: I’m not entirely clear if all the V’O2-accel calibrations used in this study were derived from respirometry-treadmill studies undertaken in previous studies, or the present study. Don’t such calibrations already exist for pumas from previous work? If new calibrations were constructed in the current study, why, and what animal numbers were used, what speeds were used, and how do the weights of those animals compare to those in the field? And how physically fit were the treadmill animals? If long-term captive, perhaps less fit than individuals in the field? Or were they the same individuals?

L140: GMT in full first time

L142: Check grammar

L144-146: Unclear

L160: ‘Energy demand’ seems a slightly odd phrase here. Would e.g. ‘energy expenditure’ be better?

L160: In the Methods you don’t go on to explain who you quantified either maneuvering or the impact of landscape features. For example, do your treadmill calibrations account for going uphill, or going downhill? It is unclear that the relationship between V’O2 and angle holds well at all (see Halsey L, Shepard E, Hulston C, Venables M, White C, et al. (2008) Acceleration versus heart rate for estimating energy expenditure and speed during locomotion in animals: tests with an easy model species, Homo sapiens. Zoology 111: 231-241.)

L163: The smoothing is done during the derivation of ODBA, isn’t it? I.e. rather than ODBA itself being smoothed. Also, I suspect that the optimal smoothing window will vary with the animal and the primary movement being recorded, so 2 s might not be ideal here? Unlikely to be best for both species?

L177-179: This needs a bit more explaining – what video cameras and where? How did you can video of the start of the run by the pumas? Some further detail given in the next sentence but this initial sentence still needs working on.

L201: ‘elevation change’ – I still don’t know what you are doing with this variable.

L210: ‘Compared to the initial escape, each puma’s second flight’ – I’m not sure this has been well explained previously.

L212-213: As mentioned earlier, this issue (is tortuosity the same as maneuvrability?) needs to be explained earlier.

L219 – jumping into trees: how do you know this?

L219-221: ‘fleeing up steep, wooded hillsides were all used repeatedly to increase escape distance and postpone being overtaken’ – this risks being anecdote. Could you, for example, quantify that when certain escape behaviours are undertaken, distance between puma and hounds temporarily increases?

L222-226: I’m not sure that chase and pursuit have been sufficiently clearly defined earlier on such that e.g. these sentences are easy to understand.

L247: ‘maximum path deviation’?

L253: why work with mass-specific numbers? We know that typically metabolic rate does not scale with body mass unitarily. How about dividing whole animal values through by 1^0.75 rather than 1^1?

L271: but if tortuosity is the same as maneuvrability (if so, then one of these two words should probably be excised from the manuscript) then Table 3 indicates no significant differences in this variable between species.

L312: typo

The Discussion should perhaps consider the results and figures a little more. Much of this section is about the implications of puma and hound energetics in relation to habitat change etc. But, for example, what about the fact that in chase four it appears the puma went well above their aerobic threshold? Does this happen to relate to the occurrences during this particular hunt? (E.g. did the animal behave differently because it got worn out?). Not much is said about Figure 4.

Table 3: Tortuosity must have units.

Figure 4 caption: ‘group centroid’ needs explaining.

Experimental design

Original work (though clarity is needed on whether the treadmill calibrations were all done as part of the current manuscript). Well defined hypotheses early on. Rigorous interrogation of the data. Methods mostly clear but a few bits need expanding - see comments provided in section 1.

Validity of the findings

Very novel, and entirely valid as far as I can tell. Sample size small but that's understandable here, and the results are still sufficiently robust in my view. Stats seems fine. Conclusions good though I feel the Discussion could attend a bit more to the study's findings - see my comment on this provided in Section 1.

Additional comments

An exciting study - 'cool' animals doing cool things. A clash of the titans, quantified!

It is not easy to make such complex behavioural interactions clear to the reader, and to quantify them so effectively.

Some 'official' overall comments below:

This is an exciting paper on what goes on during a clash of two charismatic big-hunt species. The authors have done a good job of turning highly complicated chase sequences into understandable and digestible nuggets of information about the contrasts, and similarities, between these animals. A larger sample size would naturally be even more elucidating, however it is understandable that for such a study N is relatively low, and is still tremendously insightful.

My only gripes are that some information is not sufficiently explained (see specific comments below) and that I’m unclear if the energetics calibrations the authors apply account for the big energy costs associated with moving up or down hill (see e.g. Halsey LG, White CR (2017) A different angle: comparative analyses of whole-animal transport costs running uphill. Journal of Experimental Biology 220: 161-166.).

---

## Round 0.2 · accepted · Accept

Following satisfactory revision and rebuttal, I now consider the manuscript suitable for publication.